# Advanced Silicon Modeling of Native Mitral Valve Physiology: A New Standard for Device and Procedure Testing

**DOI:** 10.3390/bioengineering12040397

**Published:** 2025-04-07

**Authors:** Katell Delanoë, Erwan Salaun, Régis Rieu, Nancy Côté, Philippe Pibarot, Viktória Stanová

**Affiliations:** 1Institut Universitaire de Cardiologie et de Pneumologie de Québec—Université Laval, Québec, QC G1V 4G5, Canada; katell.delanoe.1@ulaval.ca (K.D.); erwan.salaun.1@ulaval.ca (E.S.); nancy.cote@criucpq.ulaval.ca (N.C.); philippe.pibarot@med.ulaval.ca (P.P.); 2Faculté des Sciences Médicales et Paramédicales, Aix-Marseille Université, LBA UMR T24, 13015 Marseille, France; regis.rieu@univ-amu.fr

**Keywords:** mitral valve, in vitro, biomechanics, silicon, modeling, 3D printing

## Abstract

Mitral valve regurgitation is among the most prevalent valvular heart diseases and increases with age. Percutaneous therapy has emerged for the management of mitral regurgitation in high surgical risk patients. However, the long-term consequences of these interventions are still not fully understood due to their novelty and the difficulty of developing a strategy specific to the patient’s anatomy and/or pathology. To optimize these outcomes, an in vitro patient-specific approach could provide important insights for the most suitable strategy to use according to the patient profile. To ensure the reliability of this in vitro approach, the aim of this study was to reproduce the physiological behavior of the healthy native mitral valve for future applications. To do so, different silicon combinations reproducing the physiological anatomy of a healthy mitral valve were developed and tested under physiological hemodynamic conditions in a cardiac simulator. The hemodynamic and biomechanical behaviors of each mitral valve model were analyzed and compared to the physiological values provided in the literature. This study identified EcoFlex 00-50 and DragonSkin 10 (Smooth-On Inc., Easton, PA, USA) as the optimal silicon combination resulting in physiological strain values and hemodynamic parameters. These findings could be useful for future patient-specific applications, helping in the optimization of percutaneous mitral valve therapy.

## 1. Introduction

Mitral valve regurgitation is one of the most common valvular diseases [1], affecting 2% of the population worldwide [2,3] and over 10% of the population aged over 70 years [4]. Characterized by a retrograde flow from the left ventricle to the left atrium during the ventricular systole, which is due to an insufficient coaptation of the mitral valve leaflet, mitral regurgitation can induce ventricular and auricular remodeling, high pulmonary arterial pressure, and ultimately leads to heart failure and death [5,6]. Until now, no pharmacotherapy has been proven to be efficient to treat this condition, and the only option is to repair or replace the failed mitral valve, generally through open-heart surgery [7,8]. However, due to high surgical risk, about half of patients with severe mitral regurgitations are denied surgical therapy [9]. To address this issue, numerous percutaneous procedures using different devices have been developed over the past decade, enabling the repair or replacement of the mitral valve [10], such as Transcatheter Edge-to-Edge Repair (TEER) [11], Annuloplasty [12], or Transcatheter Mitral Valve Replacement (TMVR) [13]. Nevertheless, due to the novelty of these procedures, the long-term hemodynamic and biomechanical outcomes remain unknown. In vitro studies can help to overcome these issues by providing a better understanding of mechanical and hemodynamic outcomes in a controlled and reproducible environment, allowing the collection of a wide spectrum of patient-specific data and testing different strategies that might be impossible to test in vivo. Indeed, previous in vitro studies have been able to report hemodynamic consequences of surgical or percutaneous treatments by using a porcine or bovine physiologic or pathophysiologic mitral valve model [14,15,16]. Nevertheless, due to the heterogeneity of mitral valve pathologies, the strategy for the treatment of mitral valve disease is shifting towards an individualized approach and could benefit from in vitro patient-specific applications. Realistic models mimicking patient-specific conditions have been created by successfully replicating the anatomical and hemodynamic properties of the patient’s mitral valve in different studies [17,18,19]. However, the models that have been created are still attempting to reproduce the physiological behavior of a healthy mitral valve correctly. The goal of this study is to enhance the understanding of both healthy and pathological mitral valves by precisely reproducing their anatomical properties and testing them under controlled conditions. In the future, these results can be used to better understand the biomechanical and hemodynamic outcomes of percutaneous intervention in a patient-specific approach and, therefore, provide important insight on procedure planning and optimization and on its long-term durability.

## 2. Materials and Methods

The present study investigates the behavior induced by 12 different custom-made silicon mitral valves tested under normal hemodynamic conditions. The silicon mitral valves were fabricated using a 3D-printed mold filled with different layers of silicon. To determine the silicon combination that results in the most physiological behavior, anatomical, hemodynamic, and biomechanic responses have been analyzed using a cardiac simulator.

### 2.1. Mitral Valve Mold

To ensure a physiological reproduction of the mitral valve anatomy, the mitral valve models were based on the anatomy of the Lifelike mitral valve (Lifelike BioTissue Inc., London, ON, Canada). While the hydrogel mitral valve is typically used by surgeons for surgical training in repairing P2 prolapse, in this study, a healthy mitral valve model previously developed by the company was used as a reference. The Lifelike valve was imaged with a desktop micro-CT scanner (NanoScan PET-CT, Mediso, Budapest, Hungary), and the high-resolution images were imported into a 3D Slicer [20] for segmentation (Figure 1a). The finalized model was imported into MeshMixer (Autodesk Inc., San Francisco, CA, USA) to create a negative mold. The mold was then 3D-printed (Figure 1b) with a Lulzbot Taz Pro Dual Extruder printer (Lulzbot, Fargo, ND, USA) using NinjaFlex material (NinjaTek 3D, Manheim, PA, USA), the elasticity of which allowed for the easy extraction of dried silicon mitral valve models.

### 2.2. Silicon Mitral Valve

To guarantee a homogenous curing of the silicon combinations (Table 1), the 3D-printed molds were covered with silicon layer by layer using a specific number of layers for each type of silicon to obtain a predefined leaflet thickness. Leaflet thickness of a healthy native mitral valve ranges from 1 to 2 mm [21], therefore three custom-made mitral valve models were created with a leaflet thickness of 1, 1.5, and 2 mm. The chordae tendinae were replicated using six de-braided strings (100% polyester, Gütterman GmbH, Gutach im Breisgau, Germany) per valve (Figure 1c) inserted within the silicon layers, ensuring the tension during the cardiac cycle. Each string was de-braided and divided into three, providing 18 points of tensile force applied on the mitral valve’s leaflets. The silicon models were then left to dry for several hours before being removed from their molds (Figure 1d).

### 2.3. Material Properties

In this study, four different silicon elastomers (i.e., EcoFlex 00-30 (EF30), EcoFlex 00-50 (EF50), DragonSkin 10 Very Fast (DS10), DragonSkin 20 (DS20) (Smooth-On Inc., Easton, PA, USA)) with different mechanical properties were used to create mitral valve models and were compared to identify which one most closely mimics the tissue of the healthy native mitral valve. Additionally, different combinations of these elastomers were used to recreate the leaflet’s microstructure, mimicking elastin and collagen fibers within the different layers of the leaflet as each layer contributes to ensure an optimal hemodynamic and mechanical environment without any abnormal hemodynamic disturbance. The different combinations are referred to in Table 1. The previously cited elastomers (EF30 and DS10) were chosen due to their previous use in the literature for in vitro mitral implications. Indeed, EF30 has already been used for its cutting and resection feeling close to those experienced by surgeons in surgical procedures [22]. Ginty et al. [17] made the mitral valve models using EcoFlex 00-30 due to its realistic leaflet flexibility and ultrasound imaging. However, in this study, gauze needed to be placed between layers to improve the tensile strength of the silicon leaflets. Another study by Premyodhin et al. [23] tested DragonSkin 10 for its flexibility and tearing behavior. According to the surgical consultant, the tearing behavior was suitable for reproducing cutting–suturing while maintaining a realistic flexibility. For the purpose of this study, EcoFlex 00-50 and DragonSkin 20 were added to the tested elastomers as they provided different Shore hardness or other material properties useful for closely matching the mitral valve properties. Young’s Modulus of each elastomer was evaluated by equi-biaxial testing with a maximal force applied of 0.5 N at a loading speed of 0.01 mm/s. Material properties found in the literature [24,25,26,27,28,29] and obtained by uniaxial and equi-biaxial testing are summarized in Table 2. Stress–strain curves obtained from material mechanical testing can be found in Appendix A.

### 2.4. In Vitro Testing

In vitro testing was performed using a left heart duplicator system [30]. This system includes anatomically shaped and deformable silicon models of the left ventricle and aorta. The left ventricle cavity is surrounded by liquid and is activated by piston pumps (Vivitro Inc., Victoria, BC, Canada) controlled using LabVIEW8.2 (National Instruments, Austin, TX, USA). Left atrium pressure is reproduced with a water column, allowing a clear visualization of the mitral leaflets during a cardiac cycle. The duplicator system is filled with a mixture of glycerin and water, which results in a fluid with the same viscosity as blood (3.8–4.0 cP). Transvalvular flow was measured with an electromagnetic flowmeter (Model 501, Carolina Medical Electronics Inc., East Bend, NC, USA, accuracy ± 1% maximum full scale) positioned immediately before the aortic valve and averaged over 100 cycles. Pressures in the left ventricle and the aortic root were recorded by micro-tip pressure catheters (Millar catheter and signal conditioning unit, Millar Instruments, Houston, TX, USA, accuracy ± 0.5% maximum full scale). Doppler echocardiographic measurements were performed using Philips iE33 (Philips, Eindhoven, The Netherlands). The transvalvular flow velocities, mean pressure gradient (*MPG*), Diastolic Filling Period (*DFP*), and mitral Velocity–Time Integral (*VTI*) were measured ten times per condition by continuous-wave Doppler. The mitral valve area (*MVA*) was estimated using the Gorlin formula [31].(1)$$MVA\left(cm^{2}\right)=\frac{CO}{0.85*37.9*\sqrt{MPG}*HR*DFP}$$

*CO* = Cardiac Output (mL/min)

*HR* = Heart Rate (bpm)

Each of the silicon mitral valves was tested under experimental conditions replicating the heart rate, the stroke volume (SV), and mean aortic pressure of a healthy individual (70 bpm, 70 mL, and 100 mmHg, respectively). To measure strain repartition and leaflet displacement of the mitral valve models, a non-contact optical 3D Digital Image Correlation method (DIC) was conducted with commercial system VIC3D (Correlated Solutions, Inc., Irmo, SC, USA). Two high-speed video cameras (1000 img/s) (FASTCAM SA3; Photron, Inc., San Diego, CA, USA), equipped with 105 mm lenses (EF 24 Reflex lenses; Sigma Corporation, Kanagawa, Japan), were used to record the leaflet motion. Finally, the mitral valve models were tattooed with black tissue dye (Killer Ink Tattoo., Liverpool, UK) in order to create a fine speckled pattern allowing the use of DIC. This method allows a direct assessment of local displacements and deformations/strains of the mitral leaflet throughout one cardiac cycle for a better comparison of the mechanical properties of the different mitral valves created in this study. Finally, evaluation of the geometric diastolic orifice area (GOA) was performed using a custom-coded Matlab program (The Mathworks Inc., Natick, MA, USA) coupled with the video of leaflet motion acquired by the high-speed cameras.

### 2.5. Statistical Analysis

Continuous variables were presented as mean values ± SD and were compared using the Student’s *t*-test (*p*-value < 0.05 considered statistically significant).

## 3. Results

### 3.1. Anatomical Properties

The first step in evaluating the characteristics of reproduced mitral valves (Figure 2) was evaluating their anatomical characteristics. The custom-made mitral valves were composed of a mitral annulus with a thickness of 4 mm, surrounding the leaflet area and replicating an elliptic shape with the antero-posterior diameter smaller than the commissural length (Figure 2a, *p* < 0.001) as observed in in vivo studies [32]. The silicon mitral valves consisted of two leaflets, anterior and posterior, each divided into three leaflet segments: A1, A2, A3, P1, P2, P3, as illustrated in Figure 2. To ensure the physiological anatomy, the anterior leaflet was longer than the posterior (Figure 2b, *p* < 0.001), with the P2 segment being longer than the P1 (*p* < 0.001) and longer than P3 (*p* < 0.001) [33]. Six de-braided strings were placed between two distinct layers of silicon, generating a simplified representation of chordae tendineae and therefore allowing the tensile applications on the free edge of the leaflet, preventing them from flailing into the left atrium during the systolic cardiac phase. Leaflet thickness was set firstly at 1.5 mm for each model, then only the valves that passed the hemodynamical properties test were replicated with 1 and 2 mm for evaluation of thickness impact on mitral valves’ hemodynamics.

### 3.2. Hemodynamic Properties

Following validation of the physiological and anatomical reproductions of native mitral valves, each silicon mitral valve model was tested under physiological conditions to study their hemodynamic behavior (Table 3). Continuous Doppler acquisition demonstrated that every valve induced a physiological response, with E and A waves, which are characteristic of healthy native mitral flow. Only one model (DS10EF30 (=V9)) had a regurgitant orifice area of 0.20 cm^2^, consistent with mild to moderate regurgitation in clinical guidelines [34]. All the other silicon mitral valves had none or trace mitral regurgitation. Based on clinical guidelines (Table 4) [35,36], three silicon models (EF30 (=V1), EF30DS10 (=V5), and EF50DS10 (=V7)) were able to reproduce healthy physiological hemodynamic conditions with a *MVA* > 4 cm^2^, a GOA > 4 cm^2^, *MPG* ≤ 2 mmHg, Vmax < 1.5 cm/s, *VTI* < 31 cm, and a DVI < 2.2 (Table 3).

### 3.3. Leaflet Thickness

Based on hemodynamic results, the silicon mitral valves V1, V5, and V7 were selected to study the influence of leaflet thickness on valve hemodynamic parameters. Three different thicknesses (1, 1.5, and 2 mm) of leaflets for each valve were tested under physiological conditions in the cardiac simulator. *MVA*, GOA, and *MPG* were analyzed for each silicon mitral valve (Figure 3, Table 5). All the silicon mitral valves induced a mitral flow profile characterized by physiological E and A flow waves, except for valve V1 at 1 mm due to its high elastic composition. As shown in Figure 3, the largest GOA and *MVA* (*p* < 0.005) for each combination of silicon mitral valve was obtained when the leaflet thickness was fixed at 1.5 mm. Similar results were found when analyzing the *MPG*; the leaflet thickness of 1.5 mm was the only one inducing a value under 2 mmHg.

### 3.4. Biomechanical Properties

To ensure the physiological reproduction of the native mitral valve, the next step was to validate the biomechanical properties of the three selected silicon mitral valves (V1, V5, and V7). Leaflet displacement evaluation is necessary to ensure that prolapse behavior is not present, even though the regurgitant orifice area has been assessed for these valves before. The leaflet prolapse is defined as the displacement of leaflet tissue into the left atrium, more than 2 mm past the mitral annular plane during the systolic cardiac phase. After the visualization of the silicon mitral valves during the mid-systolic cardiac phase (Figure 4), it appeared that the V1 valve caused a consistent A3/P3 prolapse, which prevented it from further analysis using the DIC process.

Even though V5 and V7 silicon mitral valves seemed to present a non-prolapsing behavior, the displacement past the mitral annular plane (Z displacement) of each leaflet segment was analyzed. As shown in Figure 5, the Z displacement of each leaflet segment was higher for the V5 valve compared to the V7 valve or to the Lifelike mitral valve model (*p* < 0.005). For both silicon mitral valves, the highest leaflet displacement was found in the A2 segment. However, the Z displacement was significantly higher for the V5 valve, which induced a slight leaflet prolapse (Figure 5, *p* < 0.001).

The accuracy for the DIC was 0.013 pixels. The major (E1) and minor (E2) principal strains were analyzed during the systolic cardiac phase. Positive signs in strain describe tension, and negative signs describe compression. The highest strains are mainly localized on the A3/P3 segment for the Lifelike model and on the coaptation line for the V7 valve, whereas the V5 seemed to provide a homogenous strain distribution throughout the entire leaflet surface (Figure 6). Besides the strain repartition, the strain intensity on each leaflet segment showed that the highest values of major principal strains were indeed induced by the A3/P3 segment on the Lifelike model (Figure 7) along with minor principal strain values (Figure 8). Regarding V5 and V7 silicon mitral valves, there were not any leaflet segments inducing prominent strain as suspected. Finally, average major principal strains were significantly lower on the V7 silicon whole leaflet surface compared to strains induced by the V5 and Lifelike mitral valve (Anterior Leaflet (AL): 11.3%, Posterior Leaflet (PL): 12.0%, Total: 12.0% vs. AL: 26.0%, PL: 25.0%, Total: 25.0% vs. AL: 23.0%, PL: 24.0%, Total: 24.0%, respectively, *p* < 0.001).

## 4. Discussion

In this study, we sought to create a silicon model that reproduces the physiological behavior of a healthy native mitral valve to fulfill the need for realistic models that could then be applied for testing the different types of mitral valve percutaneous procedures and devices. The hydrogel mitral valve featured great shape fidelity but would not allow a rapid patient-specific anatomy reproduction, and therefore, it might be an expensive alternative for testing percutaneous devices. Although previous studies have demonstrated the usefulness of silicon for mimicking the anatomy of the human mitral valve [18,39] or the sensation of suture feeling during mitral valve surgical training [23,40], only a few studies examined their capabilities to reproduce the hemodynamic behavior of the native mitral valve. Previous studies investigated the hemodynamic outcomes of mitral valve interventions in a pulsed duplicator system [17,19,41], but to our knowledge, the physiological behavior of a healthy mitral valve has not been successfully reproduced yet. Hence, the goal of this study was to reproduce the rheological and hemodynamic behavior of a healthy mitral valve by testing and comparing different silicons (and their combinations) used in previous literature [17,22,23] in order to provide a framework for future patient-specific studies reproducing the mitral valve behavior.

The main finding of this study is that silicon combination EF50DS10 (=V7) was able to replicate the anatomical features of a healthy mitral valve while inducing a normal physiological hemodynamic behavior. Indeed, the EF50DS10 silicon mitral valve with a thickness of 1.5 mm achieved *MVA* > 4 cm^2^, GOA > 4 cm^2^, and *MPG* ≤ 2 mmHg, which is characteristic of physiological hemodynamic function. Regarding the biomechanical performance, this silicon combination was one of the few providing a GROA < 0.20 cm^2^ and a prolapsus < 2 mm, indicating a non-pathological behavior. Finally, its systolic leaflet strain (≈10%) was comparable to strains measured in vivo [42,43]. The systolic strain induced in the silicon mitral valve was slightly higher compared to the in vivo study performed by El-Tallawi et al. [42] (AL: 11.3%, PL: 12.0%, Total: 12.0% vs. AL: 7.6%, PL: 9.3%, Total: 8.5%, respectively), but this could be explained by the non-saddle shape of the mitral annulus of the Lifelike mitral valve model and the simplified configuration of the chordae. Indeed, the complex form of the mitral annulus and the attachment of the chordae on each leaflet are known for reducing leaflet stress and strain [44,45]. While the aim of this study was focused on comparing different leaflet silicon combinations, future studies could benefit from adding more chordae to reduce the leaflet strain. In addition, reproducing the saddle-shaped annulus of the native mitral valve could also prevent leaflet prolapse and higher strains.

In summary, the purpose of the current study was to provide a physiological model of a native healthy mitral valve to respond to the deficiencies of the current mitral valve models used in in vitro experiments. This study offers a low-cost, simple, and customizable process to develop a healthy/pathological generic or patient-specific mitral valve. Although these models were initially developed for patient-specific in vitro studies, their low production cost makes them highly suitable for educational and training purposes. Aside from the one-time expense of purchasing a 3D printer, the cost of producing a silicon mitral valve is less than USD 10, making it widely accessible for different applications. Furthermore, their high material stability, reproducibility, and compatibility with imaging and sensors make them valuable for R&D, particularly in testing the compatibility of mitral transcatheter devices with MRI. Lastly, these silicon models could also aid in procedural planning to predict and prevent obstruction of the left ventricular outflow tract before mitral valve replacement. However, despite its material stability and 24-month shelf-life, the authors recommend using the silicon mitral valve within the year after its creation to assure data reliability.

While this research focused on the development of healthy mitral valves, results from the different silicon combinations tested in this study could be useful to develop and understand different mitral valve dysfunction. Indeed, even if every silicon combination was able to reproduce anatomical features of a healthy mitral valve, physiological hemodynamic function was induced only by three mitral valves (V1, V5, and V7) with a leaflet thickness of 1.5 mm. Based on these results, stenotic behavior (i.e., *MPG* > 5 mmHg) could be reproduced using the combination of V10, V11, and/or V3 (Appendix A). The silicon combination EF30 could reproduce the leaflet prolapsus resulting from an early fibroelastic deficiency (Appendix A), and varying leaflet thickness could also be interesting for inducing different pathological behavior. Indeed, the V1 silicon combination with 1 mm leaflet thickness induced a Barlow-type behavior, which represents a challenging clinical entity to manage. Therefore, it would be interesting to further investigate the appropriate silicon combinations and chordae tension to use while reproducing pathophysiological conditions. While this study sought to identify an optimal silicon combination for replicating the physiological behavior of a healthy mitral valve, further study is needed to validate the model’s accuracy under patient-matched anatomical and hemodynamic conditions. Additionally, to enhance the physiological reproduction and minimize strain discrepancies, a more precise chordae and annulus model will be developed based on patient-specific anatomy.

Finally, to further reproduce the patient-specific rheological tissue, the porosity reproduction of different pathophysiological mitral valves could be useful as it has been linked to material strength, durability, and permeability properties [46]. Indeed, as highlighted by Fatmanur Kocaman Kabil and Ahmet Yavuz, increasing the porosity of silicon rubbers enhances flexibility while reducing stiffness, which might be relevant to mimic mitral valve biomechanics. In this case, it could be interesting to recreate a silicon mitral valve with a different porosity to match the fractal dimension obtained from different mitral valve physiological states [47].

This study provides evidence of the reproduction of physiological mitral valve behavior, which could be helpful for future research focused on understanding the effects, outcomes, and long-term durability of percutaneous interventions and devices. Indeed, further investigation into the development of in vitro mitral valves could lead toward the reproduction of patient-specific interventions, leading to optimization of the planning of the interventions. Defining the connection between hemodynamic and mechanical parameters could be crucial for optimizing the efficiency of the treatment by highlighting the most suitable device to use with regard to the size, number, or type depending on the patient’s specific anatomy and disease etiology. The characterization of these optimized associations has the potential to enhance the quality of life for patients and the device’s durability while reducing the risk of surgical reintervention.

## 5. Conclusions

In conclusion, the silicon combination of EcoFlex 00-50 and DragonSkin 10 with a thickness of 1.5 mm enables the closest reproduction of the physiological, anatomic, hemodynamic, and biomechanical behavior of a healthy mitral valve. The results of this study provide important new insights on the optimal silicon combination for reproducing patient-specific physiological and pathological mitral valves undergoing percutaneous interventions, resulting in a higher level of reliability between in vitro and in vivo research. Further investigation into the behavior of silicon combinations could improve the ability to reproduce the pathophysiological behavior of the native mitral valve. Additionally, this study presents a cost-effective alternative for testing transcatheter interventional devices prior to implantation, potentially reducing the financial and procedural burden associated with single-use, sterilized devices. This progress may pave the way for a patient-specific approach leading toward an individualization of the percutaneous treatment of the mitral valve by tailoring the procedural strategy on the etiology and anatomy of the individual patient.

## Figures and Tables

**Figure 1 bioengineering-12-00397-f001:**
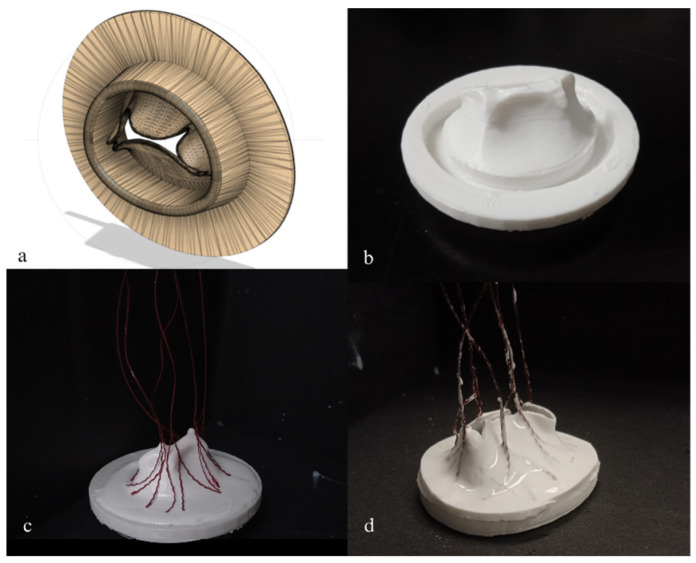
Development processus of silicon mitral valves: (**a**) Lifelike 3D model, (**b**) 3D-printed mold, (**c**) chordae added between silicon layers, (**d**) final silicon mitral valve.

**Figure 2 bioengineering-12-00397-f002:**
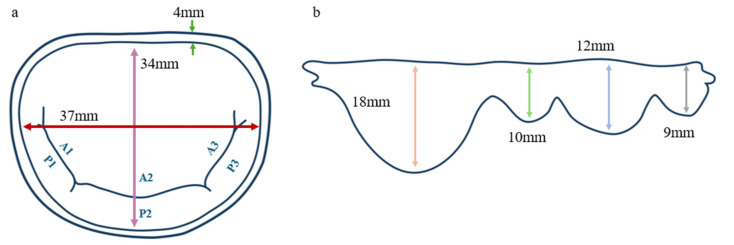
Anatomical characteristics of silicon mitral valves on (**a**) surgical view, (**b**) ventricular view. Different segments of the mitral valve anatomy were measured on different views. Surgical view (**a**): long axis diameter (red line), short axis diameter (pink line), annulus thickness (green arrow). Ventricular view (**b**): anterior leaflet length (beige arrow), P3 length (green arrow), P2 length (blue arrow), P3 length (grey arrow).

**Figure 3 bioengineering-12-00397-f003:**
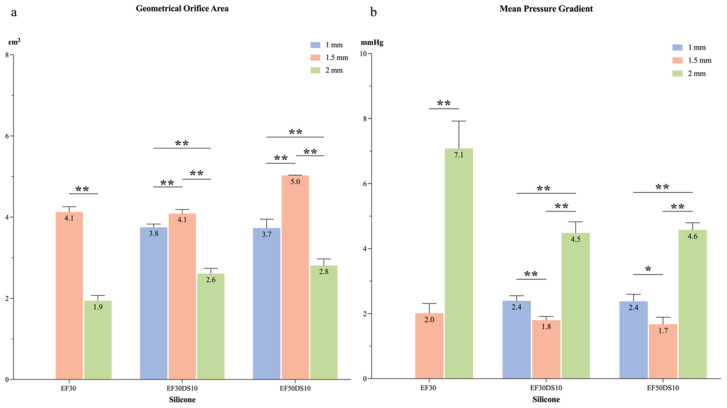
Hemodynamic parameters depending on leaflet thickness: (**a**) GOA, (**b**) *MPG*. Comparison of hemodynamic behavior induced by three chosen silicon mitral valves with three different leaflet thicknesses (EF30 = V1, EF30DS10 = V5, EF50DS10 = V7); *: *p* < 0.005, **: *p* < 0.001.

**Figure 4 bioengineering-12-00397-f004:**
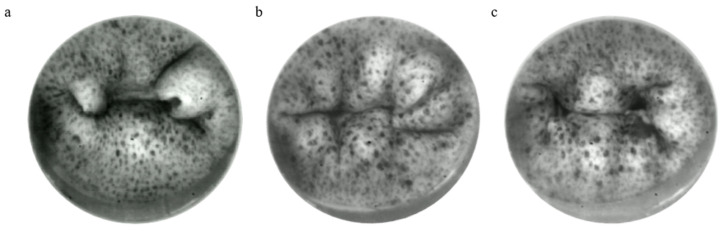
Mid-systolic closing of silicon mitral valves (**a**) V1, (**b**) V5, (**c**) V7. Three silicon mitral valves inducing the most physiological behavior were analyzed using high-speed video cameras for comparing their rheological behavior during the systolic cardiac phase.

**Figure 5 bioengineering-12-00397-f005:**
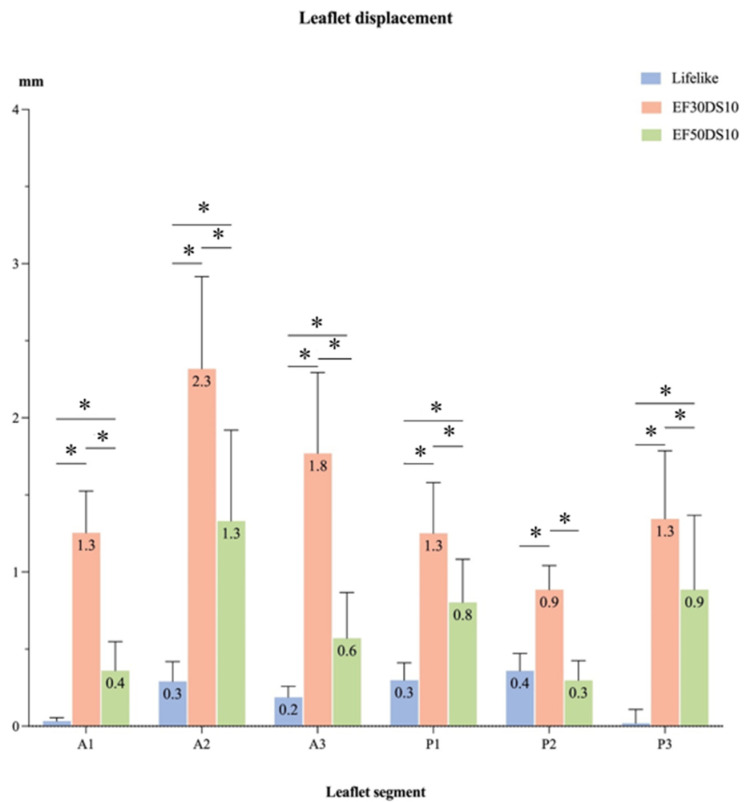
Leaflet segment displacement (mm) during the systolic phase. Leaflet Z displacement during the systolic cardiac phase represented for different regions of the mitral valve (A1, A2, A3, P1, P2, P3) (EF30DS10 = V5, EF50DS10 = V7); *: *p* < 0.001.

**Figure 6 bioengineering-12-00397-f006:**
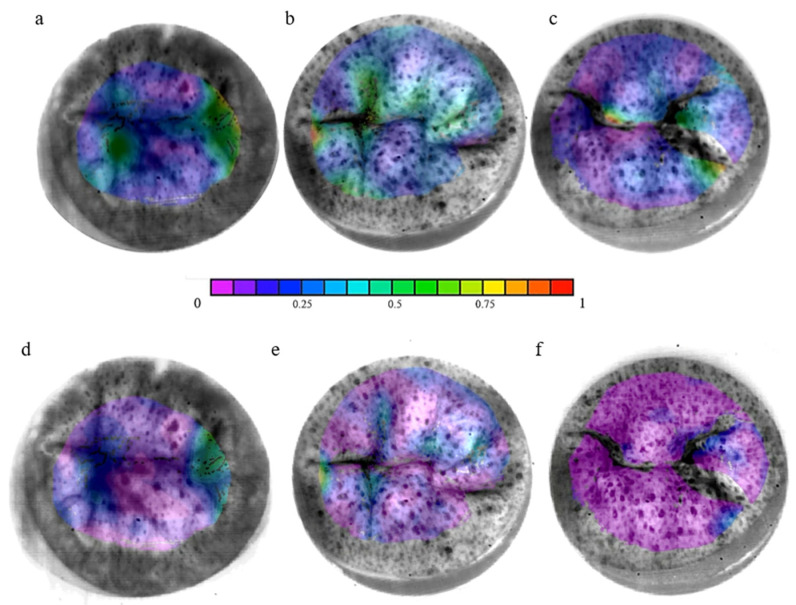
Major (E1) and minor (E2) principal strain of silicon mitral valves during the systolic phase: (**a**) Lifelike E1, (**b**) V5 E1, (**c**) V7 E1, (**d**) Lifelike E2, (**e**) V5 E2, (**f**) V7 E2. Strain field on the leaflet surface of the two most physiological silicon mitral valves with comparison to the model hydrogel mitral valve. Major and minor strains were analyzed by using the DIC technique during the systolic cardiac phase (V5 = EF30DS10, V7 = EF50DS10).

**Figure 7 bioengineering-12-00397-f007:**
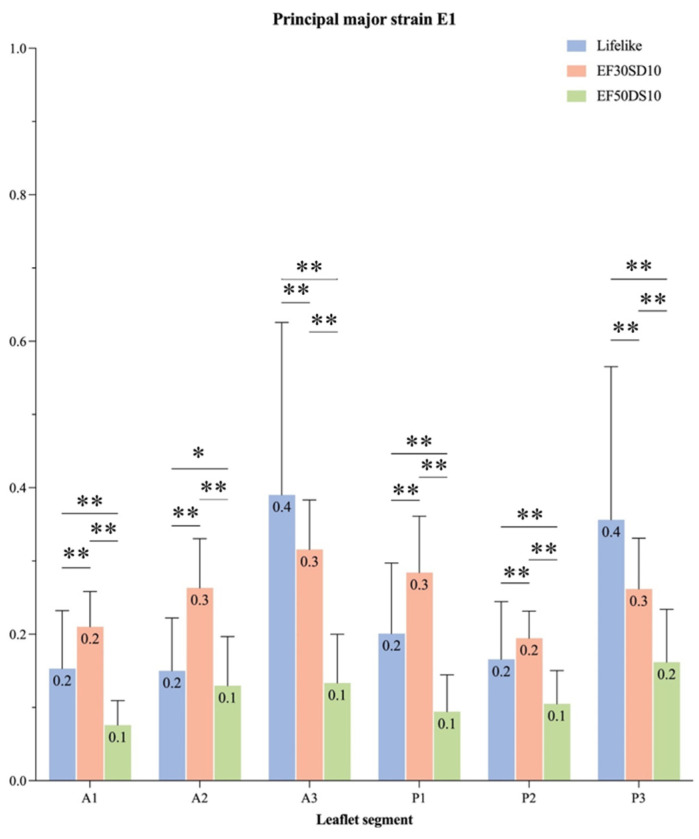
Major principal strain (E1) of each leaflet segment of silicon mitral valves during the systolic phase. Principal major strain (E1) during the systolic cardiac phase represented for different regions of the mitral valve (A1, A2, A3, P1, P2, P3) (EF30DS10 = V5, EF50DS10 = V7); *: *p* < 0.005, **: *p* < 0.001.

**Figure 8 bioengineering-12-00397-f008:**
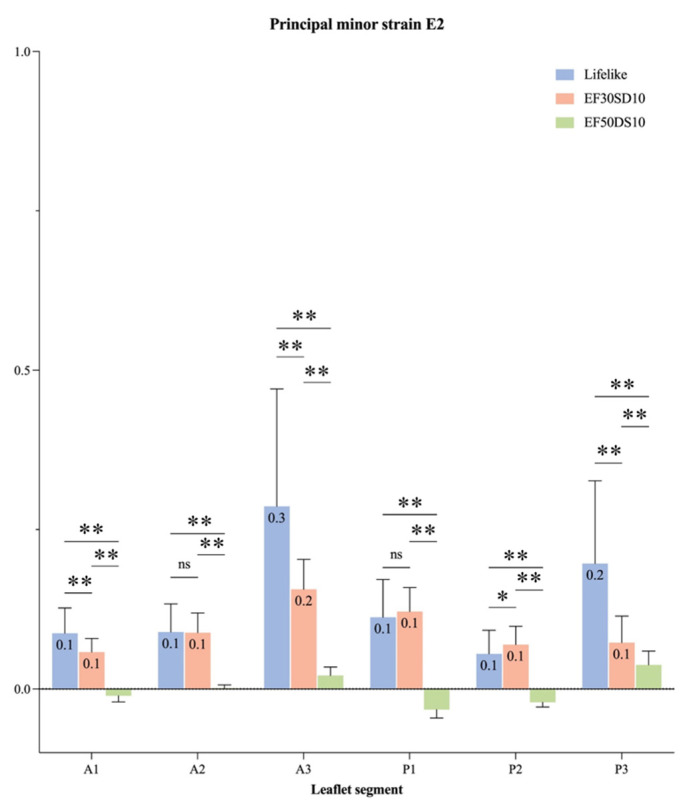
Minor principal strain (E2) of each leaflet segment of silicon mitral valves during the systolic phase. Principal minor strain (E2) during the systolic cardiac phase represented for different regions of the mitral valve (A1, A2, A3, P1, P2, P3) (EF30DS10 = V5, EF50DS10 = V7); *: *p* < 0.005, **: *p* < 0.001; ns: *p* > 0.05.

**Table 1 bioengineering-12-00397-t001:** Silicon combinations used for the development of custom-made mitral valves.

Valve	External Silicon Layers	Internal Silicon Layers
V1	EcoFlex 00-30	
V2	EcoFlex 00-50	
V3	DragonSkin 10 Very Fast	
V4	DragonSkin 20	
V5	EcoFlex 00-30	DragonSkin 10 Very Fast
V6	EcoFlex 00-30	DragonSkin 20
V7	EcoFlex 00-50	DragonSkin 10 Very Fast
V8	EcoFlex 00-50	DragonSkin 20
V9	DragonSkin 10 Very Fast	EcoFlex 00-30
V10	DragonSkin 10 Very Fast	EcoFlex 00-50
V11	DragonSkin 20	EcoFlex 00-30
V12	DragonSkin 20	EcoFlex 00-50

**Table 2 bioengineering-12-00397-t002:** Material properties of tested silicons.

Material	Young’s Modulus (MPa)	Ultimate Tensile Strength (MPa)	Elongation at Break (%)	Failure Strength (kN/m)	Shore Hardness (00-A)
EcoFlex 00-30	0.33	1.2	835	6.66	00-23
EcoFlex 00-50	0.33	1.7	860	8.77	00-35
DragonSkin 10 Very Fast	1.04	3.28	1000	17.9	10A
DragonSkin 20	3.84	3.79	620	21	20A
Native Mitral Leaflet (circ.)	0.02–10.2	______	______	0.981	______
Native Mitral Leaflet (rad.)	0.02–2.1	______	______	0.657	______

**Table 3 bioengineering-12-00397-t003:** Hemodynamic parameters induced by the silicon mitral valve.

Valve	*MVA* (cm^2^)	Peak GOA (cm^2^)	*MPG* (mmHg)	Vmax (cm/s)	*VTI* (cm)	DVI	GROA (cm^2^)
Lifelike	4.0 ± 0.2	4.7 ± 0.1	2.7 ± 0.4	1.5 ± 0.1	35.2 ± 2.8	1.0 ± 0.1	0.00 ± 0.0
V1	4.6 ± 0.3	4.1 ± 0.1	2.0 ± 0.3	1.3 ± 0.1	29.4 ± 2.6	0.8 ± 0.1	0.02 ± 0.0
V2	3.6 ± 0.3	3.2 ± 0.2	3.4 ± 0.5	1.6 ± 0.1	33.7 ± 2.5	1.0 ± 0.1	0.02 ± 0.0
V3	1.8 ± 0.1	1.4 ± 0.1	12.7 ± 0.8	3.2 ± 0.1	66.5 ± 3.0	1.9 ± 0.1	0.03 ± 0.0
V4	3.8 ± 0.2	3.2 ± 0.1	2.9 ± 0.3	1.5 ± 0.1	30.7 ± 2.3	0.9 ± 0.1	0.03 ± 0.0
V5	4.8 ± 0.1	4.0 ± 0.2	1.8 ± 0.1	1.2 ± 0.0	29.3 ± 0.8	0.8 ± 0.0	0.06 ± 0.0
V6	3.3 ± 0.1	3.0 ± 0.1	4.1 ± 0.4	1.9 ± 0.1	37.2 ± 2.4	1.1 ± 0.1	0.00 ± 0.0
V7	5.1 ± 0.4	5.0 ± 0.2	1.7 ± 0.2	1.2 ± 0.1	28.2 ± 2.1	0.8 ± 0.1	0.07± 0.0
V8	4.2 ± 0.2	3.7 ± 0.1	2.5 ± 0.2	1.4 ± 0.1	30.4 ± 1.9	0.9 ± 0.1	0.06 ± 0.0
V9	3.9 ± 0.2	4.3 ± 0.3	2.8 ± 0.3	1.7 ± 0.1	37.3 ± 2.1	1.1 ± 0.1	0.20 ± 0.0
V10	2.3 ± 0.1	2.9 ± 0.1	8.0 ± 0.9	2.7 ± 0.1	53.1 ± 3.2	1.5 ± 0.1	0.00 ± 0.0
V11	2.5 ± 0.1	2.6 ± 0.1	7.0 ± 0.5	2.5 ± 0.1	52.2 ± 2.9	1.5 ± 0.1	0.00 ± 0.0
V12	3.7 ± 0.2	3.7 ± 0.1	3.2 ± 0.3	1.7 ± 0.1	32.9 ± 2.0	0.9 ± 0.1	0.04 ± 0.0

*MVA*: Mitral Valve Area; GOA: Geometric Orifice Area; *MPG*: Mean Pressure Gradient; Vmax: Vitesse Maximale; *VTI*: Velocity–Time Integral; DVI: Doppler Velocity Index; GROA: Geometric Regurgitant Orifice Area.

**Table 4 bioengineering-12-00397-t004:** Hemodynamic parameters referred to in clinical guidelines [34,37,38].

	*MPG* (mmHg)	*MVA* (cm^2^)	GROA (cm^2^)
Healthy	≤3	4–6	<0.20
Mild to moderate	<5	>1.5	0.20–0.29
Severe	5–10	<1.5	0.30–0.39

*MVA*: Mitral Valve Area; GROA: Geometric Regurgitant Orifice Area; *MPG*: Mean Pressure Gradient.

**Table 5 bioengineering-12-00397-t005:** Hemodynamic parameters induced by different leaflet thickness of three silicon mitral valves. N/A: Not Applicable.

Valve	Leaflet Thickness (mm)	*MVA* (cm^2^)	GOA (cm^2^)	*MPG* (mmHg)
V1	1	N/A		
	1.5	4.6 ± 0.3	4.1 ± 0.1	2.0 ± 0.3
	2	1.8 ± 0.1	1.9 ± 0.1	7.1 ± 0.8
V5	1	3.0 ± 0.2	3.8 ± 0.2	2.4 ± 0.2
	1.5	4.8 ± 0.1	4.0 ± 0.2	1.8 ± 0.1
	2	2.2 ± 0.1	2.6 ± 0.1	4.5 ± 0.3
V7	1	3.7 ± 0.2	3.7 ± 0.2	2.4 ± 0.2
	1.5	5.1 ± 0.4	5.0 ± 0.2	1.7 ± 0.2
	2	2.1 ± 0.1	2.8 ± 0.2	4.6 ± 0.2

*MVA*: Mitral Valve Area; GOA: Geometric Orifice Area; *MPG*: Mean Pressure Gradient.

## Data Availability

The raw data supporting the conclusions of this article will be made available by the authors on request.

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
