# Peer review of "Advanced Silicon Modeling of Native Mitral Valve Physiology: A New Standard for Device and Procedure Testing"

_bioengineering, 2025, doi:10.3390/bioengineering12040397_

Round 1

Reviewer 1 Report

Comments and Suggestions for Authors

This paper explores the development of silicon-based models to accurately simulate the physiological behavior of the healthy native mitral valve. The authors utilized 3D-printed molds and tested various silicone combinations under controlled hemodynamic and biomechanical conditions to identify the optimal material properties. The study highlights EcoFlex 00-50 and DragonSkin 10 as the optimal combination for achieving realistic anatomical, hemodynamic, and biomechanical parameters. The results indicate potential applications in patient-specific interventions to enhance percutaneous mitral valve therapies and optimize procedure planning.

1. Although the in vitro approach provides valuable insights, there is no comparison with in vivo studies to validate the model's accuracy under biological conditions. Integrating in vivo testing would strengthen the conclusions.

2.The model's chordae tendineae and mitral annulus lack the complex geometries observed in vivo. Future iterations could incorporate these details to reduce strain discrepancies and improve biomechanical fidelity.

3. The paper emphasizes healthy valve modeling, but its potential for pathological and patient-specific scenarios is only briefly addressed. A more robust exploration of pathological conditions would enhance clinical applicability.

4. The authors are required to discuss the significance of porosity in relation to the properties of the Mitral valve, similar to the approach taken in the publication "fractal-based approach to the mechanical properties of recycled aggregate concretes". This would provide a more comprehensive understanding of the factors influencing the mitral valve's characteristics.

Author Response

Summary

We would like to thank the reviewers for their valuable comments, which have allowed us to improve our manuscript. In response to the feedback received, we have revised the article accordingly and provided detailed responses in this document.

We include below a detailed answer to each comment in a point-by-point fashion. The responses to the concerns raised by reviewers are below and are coded as follows: a) Comments from editors or reviewers are shown as bold text; b) Our responses are shown as italic text. The changes in the revised paper can be found in “Original Revised Paper.docx” and in the present document and are highlighted in green.

We hope these revisions align with your expectations and prove satisfactory. Nevertheless, we remain open to any further modifications that may contribute to the continued improvement of our manuscript.

Reviewer 1:

This paper explores the development of silicon-based models to accurately simulate the physiological behavior of the healthy native mitral valve. The authors utilized 3D-printed molds and tested various silicone combinations under controlled hemodynamic and biomechanical conditions to identify the optimal material properties. The study highlights EcoFlex 00-50 and DragonSkin 10 as the optimal combination for achieving realistic anatomical, hemodynamic, and biomechanical parameters. The results indicate potential applications in patient-specific interventions to enhance percutaneous mitral valve therapies and optimize procedure planning.

Point-by-point response to Comments and Suggestions for Authors

Comments 1: Although the in vitro approach provides valuable insights, there is no comparison with in vivo studies to validate the model's accuracy under biological conditions. Integrating in vivo testing would strengthen the conclusions.

We would like to thank the reviewer for this comment and agree with their statement that an in vivo comparison would have strengthened the conclusion. However, as this was a preliminary study, the geometry was based on an already existing “healthy” Lifelike valve model (BioTissue Inc., ON, CA) which is a hydrogel valve certified for surgeon training and inducing physiological behavior. Despite the fact that we compared the results of our study to the Lifelike valve, Ginty et al (2017) recreated and compared successfully in vitro vs in vivo patient-specific mitral valve using EcoFlex 00-50 and gauze patches, demonstrating the potential of the silicone mitral valve. Furthermore, the main purpose of this preliminary study was to establish the most-suited silicon combination for reproducing physiological behavior. The next step for us will be to validate the patient-specific reproduction using in vivo images and their respective hemodynamic parameters, but this is a lengthy process as the use of in vivo images requires ethical committee approval, that is a lengthy process. To address this issue, a sentence has been added in the discussion (line 347-350).

Lines 347-350 : “While this study sought to identify an optimal silicone combination for replicating the physiological behavior of a healthy mitral valve, further study is needed to validate the model’s accuracy under patient-matched anatomical and hemodynamic conditions”.

Comments 2: The model's chordae tendineae and mitral annulus lack the complex geometries observed in vivo. Future iterations could incorporate these details to reduce strain discrepancies and improve biomechanical fidelity.

We would like to thank the reviewer for this comment and agree with their statement that mitral annulus and chordae complexity would have to be enhanced in future iterations. As this was a preliminary study focused on choosing a material/silicone combination that could replicate the mitral valve physiological hemodynamic behavior, we decided to start with a simplified but still accurate representation of the mitral valve. Furthermore, the Lifelike model has a flat annulus shape, and we made the choice to reproduce its exact geometry to reduce the outcomes of variability and ensure that the differences in hemodynamic results would only be induced by the silicon combination. We are currently working on a model integrating a more complex subvalvular apparel which would be used in future experiments. To address this concern a sentence has been added to the discussion section (line 355-357) in addition to the previous paragraph detailing this limitation (line 308-318).

Lines 355-357: “Additionally, to enhance the physiological reproduction and minimize strain discrepancies, a more precise chordae and annulus model will be developed based on patient-specific anatomy.”

Lines 308-318 : “The systolic strain induced in the silicon mitral valve was slightly higher compared to in vivo study performed by El-Tallawi et al (AL: 11.3%, PL:12.0%, Total: 12.0% vs. AL: 7.6%, PL: 9.3%, Total: 8.5% respectively), but this could be explained by the non-saddle shape of the mitral annulus of the Lifelike mitral valve model and the simplified configuration of the chordae. Indeed, the complex form of the mitral annulus and the attachment of the chordae on each leaflet are known for reducing leaflet stress and strain [44, 45]. While the aim of this study was focused on comparing different leaflet silicon combinations, future studies could benefit from adding more chordae to reduce the leaflet strain. In addition, reproducing the saddle-shaped annulus of the native mitral valve could also prevent leaflet prolapse and higher strains.”

Comments 3: The paper emphasizes healthy valve modeling, but its potential for pathological and patient-specific scenarios is only briefly addressed. A more robust exploration of pathological conditions would enhance clinical applicability.

We would like to thank the reviewer for their comment. This issue has been addressed, and the paragraph discussing pathological conditions has been improved with supplementary images which have been incorporated for a visual comparison of a barlow and a fibrotic mitral valve with a silicon combination inducing a similar morphological behavior (line 340 & 342).

Comments 4 : The authors are required to discuss the significance of porosity in relation to the properties of the Mitral valve, similar to the approach taken in the publication "fractal-based approach to the mechanical properties of recycled aggregate concretes". This would provide a more comprehensive understanding of the factors influencing the mitral valve's characteristics.

We would like to thank the reviewer for their insightful comment. Based on the publication "fractal-based approach to the mechanical properties of recycled aggregate concretes", porosity was shown to have a crucial influence on material strength, durability, and fluid transport properties, which are also key factors in the functioning of biological tissues like the mitral valve. Indeed, as the mitral valve is composed of a fibrous and porous extracellular matrix, it certainly exhibits similar dependencies. The fractal model applied in the concrete study suggests that porosity dictates diffusion and permeability properties, which could be analogous to fluid flow and tissue hydration in the mitral valve. Future work exploring the fractal dimension of mitral valve in different physiological states could help us understand the link between porosity and valvular dysfunction.

The silicone rubbers used in our study were non-porous with negligible shrinkage after curing and different stretchability as shown by uni and bi-axial testing. However, as highlighted by Fatmanur Kocaman Kabil and Ahmet Yavuz in the publication “Influence of the pore size on optical and

mechanical properties of ecoflex sponges”, increasing porosity of silicon rubbers enhances flexibility while reducing stiffness which might be relevant to mimics mitral valve biomechanics. In this case, it could be interesting to recreate silicon mitral valve with different porosity to match the fractal dimension obtained from different mitral valve physiological state. To discuss this matter, a paragraph has been added to the discussion paragraph (line 353-360).

Lines 353-360 : “Finally, to further reproduce the patient-specific rheological tissue, porosity reproduction of different pathophysiological mitral valve could be useful as it has been linked to material strength, durability and permeability properties [46]. Indeed, as highlighted by Fatmanur Kocaman Kabil and Ahmet Yavuz, increasing porosity of silicon rubbers enhances flexibility while reducing stiffness which might be relevant to mimics mitral valve biomechanics. In this case, it could be interesting to recreate silicon mitral valve with different porosity to match the fractal dimension obtained from different mitral valve physiological states[47].”

Reviewer 2 Report

Comments and Suggestions for Authors

In the study, the authors reported evidence regarding a silicon mitral valve model that reproduces the physiological behavior of a healthy native mitral valve to fulfill the need for realistic models. The authors reported that "silicon combination EF50DS10 (=V7) was able to replicate the anatomical features of a healthy mitral valve while inducing a normal
physiological hemodynamic behavior". The findings are impressive and practically useful. However, I woulf like to make some comments regarding the findings interpretation.

  1. The authors are welcome to extensively report the benefit of model beyond the physiological characteristics, i.e. industrial approach to creat a proof with high material stability, reproducing the model with low economic burden, etc.
  2. Please, add the economic burden in the conclusive part.

Author Response

Summary

We would like to thank the reviewers for their valuable comments, which have allowed us to improve our manuscript. In response to the feedback received, we have revised the article accordingly and provided detailed responses in this document.

We include below a detailed answer to each comment in a point-by-point fashion. The responses to the concerns raised by reviewers are below and are coded as follows: a) Comments from editors or reviewers are shown as bold text; b) Our responses are shown as italic text. The changes in the revised paper can be found in “Original Revised Paper.docx” and in the present document and are highlighted in green.

We hope these revisions align with your expectations and prove satisfactory. Nevertheless, we remain open to any further modifications that may contribute to the continued improvement of our manuscript.

Point-by-point response to Comments and Suggestions for Authors

In the study, the authors reported evidence regarding a silicon mitral valve model that reproduces the physiological behavior of a healthy native mitral valve to fulfill the need for realistic models. The authors reported that "silicon combination EF50DS10 (=V7) was able to replicate the anatomical features of a healthy mitral valve while inducing a normal
physiological hemodynamic behavior". The findings are impressive and practically useful. However, I woulf like to make some comments regarding the findings interpretation.

Comments 1: The authors are welcome to extensively report the benefit of model beyond the physiological characteristics, i.e. industrial approach to create a proof with high material stability, reproducing the model with low economic burden, etc.

We would like to thank the reviewers for their comment and have discussed further the range of possibilies when using these silicon mitral valves (lines 322-331).

Lines 322-331: “Although these models were initially developed for patient-specific in vitro studies, their low production cost makes them highly suitable for educational and training purposes. Aside from the one-time expense of purchasing a 3D printer, the cost of producing a silicone mitral valve is less than $10, making it widely accessible for different applications. Furthermore, their high material stability, reproducibility, and compatibility with imaging and sensors make them valuable for R&D, particularly in testing the compatibility of mitral transcatheter devices with MRI. Lastly, these silicone models could also aid in procedural planning to predict and prevent obstruction of the left ventricular outflow tract before mitral valve replacement.”

Comments 2: Please, add the economic burden in the conclusive part

We would like to thank the reviewers for their comments and have added a sentence about the economic burden in the conclusive part (lines 380-383).

Lines 380-383 : “Additionally, this study presents a cost-effective alternative for testing transcatheter interventional devices prior to implantation, potentially reducing the financial and procedural burden associated with single-use, sterilized devices.”
